# Update on Efficacy of Conservative Treatments for Carpal Tunnel Syndrome

**DOI:** 10.3390/jcm11040950

**Published:** 2022-02-11

**Authors:** Teemu Karjalanen, Saara Raatikainen, Kati Jaatinen, Vieda Lusa

**Affiliations:** 1Department of Hand and Micosurgery, Tampere University Hospital, 33521 Tampere, Finland; 2Monash Department of Clinical Epidemiology, Cabrini Institute, Department of Epidemiology and Preventive Medicine, School of Public Health and Preventive Medicine, Monash University, Malvern, Melbourne 3144, Australia; 3Musculoskeletal and Plastic Surgery Department, University of Helsinki and Helsinki University Hospital, 00290 Helsinki, Finland; saara.raatikainen@hus.fi; 4Central Finland Healthcare District, 40620 Jyväskylä, Finland; kati.jaatinen@ksshp.fi (K.J.); vieda.lusa@ksshp.fi (V.L.)

**Keywords:** carpal tunnel syndrome, ESWT, corticosteroid, gabapentin, Kinesio taping, orthoses, platelet-rich plasma, neurodynamic techniques, ultrasound, splint

## Abstract

Carpal tunnel syndrome (CTS) is the most common upper extremity compression neuropathy. Non-operative interventions are usually the first-line treatments, and surgery is reserved for those that do not achieve a satisfactory symptom state by non-operative means. This narrative review summarizes the current evidence regarding the efficacy of orthoses, corticosteroid injections, platelet-rich plasma injections, Kinesio taping, neurodynamic techniques, gabapentin, therapeutic ultrasound, and extracorporeal shockwave therapy in people with CTS. While many trials suggest small short-term benefits, rigorous evidence of long-term patient-important benefits is limited. To improve the utility of healthcare resources, research in this area should focus on establishing efficacy of each treatment instead of comparing various treatments with uncertain benefits.

## 1. Introduction

Carpal tunnel syndrome (CTS) is a common compression neuropathy of the median nerve at the wrist level where the nerve passes through the carpal tunnel together with the finger flexor tendons. A Swedish study estimated a prevalence of 4% in the general population based on a random sample of 3000 people [1]. Among workers, it may be up to twice as common. Pooled data from six different studies (*n* = 4321) suggested a prevalence of 8% among workers [2].

CTS is more common in females compared to males and often occurs during pregnancy implying that hormonal factors are involved in the development. Other risk factors include diabetes [3] and obesity [4,5] suggesting that metabolic factors also play a role, possibly via vascular mechanisms. Furthermore, genetic factors, rheumatoid arthritis, distal radius fracture and wrist osteoarthritis predispose to CTS [6,7,8].

Although surgery may provide more effective and durable symptom relief [9], most people with mild to moderate symptoms are initially treated non-operatively. Several RCTs have assessed the effect of various non-operative treatments in people with CTS, but often studies compare one treatment with another not addressing the question whether the interventions are better than doing nothing, i.e., their efficacy. In this narrative review, we will provide up-to-date data regarding the efficacy of orthoses, corticosteroid injections, platelet-rich plasma injections, Kinesio taping, neurodynamic techniques, gabapentin, therapeutic ultrasound, and ESWT for CTS.

## 2. Wait and See

### 2.1. How Wait and See Might Work

Many symptoms fluctuate, and CTS symptoms are no exception. People tend to seek care when symptoms are at their worst, and regression towards mean explains some proportion of the improvement at the follow-up. The symptoms may also improve if the inflammatory changes in the flexor tendon paratenon subside spontaneously.

### 2.2. Does Wait and See Work?

Evidence from observational studies indicates that the wait and see strategy can be used in people with mild and stable CTS symptoms. The spontaneous improvement rate for clinical symptom and functional outcomes varies from 23–40%, and these rates seem to be rather steady across studies irrespective of the severity of the condition [10,11,12]. Electrophysiological outcomes improve spontaneously in between 15 to 27% of people [10,11,12,13].

Many, 40–62%, of untreated CTS cases seem to stay unchanged in terms of clinical symptoms and subjective outcomes in 10 months to 2-year follow-ups [10,11,12]. Similar rates, 50–67%, are reported for electrophysiological findings [10,11,12]. The symptoms seem to predict CTS persistence at one year better than nerve conduction tests alone [14].

The deterioration rate for symptoms and hand function ranges between 4–35% without considering the severity of electrophysiological findings [10,11,12,13]. Electrophysiological findings seem to become worse in approximately 4–16% of people during one to several year follow-ups [10,11,12,13].

## 3. Orthoses

### 3.1. How Orthoses Might Work

Orthoses are a relatively inexpensive intervention that requires no regular follow-up visits. Studies measuring carpal tunnel pressure suggest that the pressure is elevated both at wrist flexion and extension [15,16,17]. Consequently, splinting the wrist at a neutral position could decrease the exposure to the elevated pressure and alleviate symptoms arising from ischemia. Moreover, it has been hypothesized that extension of the metacarpophalangeal joints could lower the pressure by moving the lumbrical muscles away from the carpal tunnel [18,19].

### 3.2. What Is the Optimal Treatment Strategy?

Several studies compare various splint designs [20]. These comparisons yield small differences between specific types of splints (commercial and/or custom-made), but the results from these studies are poorly applicable due to various splint designs used in the studies. One study found that extending the orthosis to immobilize MP-joints resulted in statistically significant yet clinically unimportant benefit in pain (1.2 points on a 0 to 10 scale) and DASH score (2.7 points on a 0 to 100 scale) at six weeks [19].

The current evidence does not indicate whether orthoses should be worn only during night time or full-time [21]. Furthermore, the optimal duration of the intervention is unclear. One study found benefit in symptoms and function at six months follow-up compared with six weeks [22], while another suggested there is no clinically important difference between six weeks and three months of use [23].

### 3.3. Do Orthoses Work?

Limited evidence supports small benefits from orthoses. Eight studies compare orthoses with no treatment or sham treatment [18,24,25,26,27,28,29,30]. Synthesis of the evidence suggests that the benefit from splinting at short-term (<3 months) may be little. Most studies find no benefit or clinically unimportant benefit ranging between 0.1 and 1.1 points on Boston Carpal Tunnel Questionnaire (BCTQ) at short term (>3 months) and 0.3 to 0.9 points at long-term [18,24,25,26,27,28,30]. However, shortcomings in methods, unexplained heterogeneity between studies and small sample sizes (imprecise treatment effect estimates) limit the interpretation, and particularly long-term benefits are unclear because limited data exists beyond 6 months.

Both short-term and long-term benefits are likely smaller than what people consider important, i.e., notice in daily life. A wide variety of possible Minimal Important Difference (MID) values for BCTQ symptoms severity score have been proposed, ranging from 0.16 to 1.45 points [31]. Based on the typical standard deviation (0.4 to 0.8), the value of 0.16 seems unreasonably low while the highest value seems unreasonably high. The scale of BCTQ is 1 to 5, a higher score indicating a worse outcome. Choosing a 10% cut off of the scale (i.e., 0.4 points) would mean that the effect of orthoses could be perceived by people with CTS but more research is needed in this area.

Regarding harms, 18% of subjects using orthoses reported some adverse effects (typically difficulty falling into sleep and transient paresthesia). However, these are not likely to persist after the intervention is discontinued.

More robust conclusions about the benefits of orthoses can be made once we obtain evidence from rigorous trials and understand better what the MID-value of the BCTQ in this context is. Furthermore, night-time symptom relief was not assessed in any of the studies, although orthoses are usually worn to relieve night-time symptoms. Since orthoses likely cause no long-term harm, they can be tried as a first-line treatment particularly when people are not interested in undergoing invasive interventions such as surgery, injections, or participating in supervised therapy.

## 4. Corticosteroid Injection

### 4.1. How Corticosteroid Injection Might Work

The exact mechanism of symptom relief is unknown, but the effects are believed to relate to the anti-inflammatory effects of corticosteroids. Inflammatory arthritis and osteoarthritis are risk factors for CTS, and animal models of compression neuropathy suggest an inflammatory response to ischemia indicating that inflammation may play a key role in the pathophysiological process [32]. Another possible mechanism is that corticosteroids may decrease oedema in the carpal tunnel thereby directly decreasing the pressure within it.

### 4.2. What Is the Optimal Treatment Strategy?

The difference between ultrasound (US) guided injection and landmark-based injection may be clinically unimportant, but US-guided injection can be preferred if it is available with little or no extra cost. A Recent systematic review [33] identified eight RCTs comparing ultra-sound (US) guided injection with landmark-based injection, but one of the studies is since rejected. The review found a small statistically significant benefit in the BCTQ symptoms severity score favoring US-guided injection (Standardized Mean Difference −0.4; 95% CI −0.2 to −0.7). Removing the retracted study and using mean difference as the summary measure, the difference in the BCTQ symptom severity score is 0.4 points (95% CI 0.15 to 0.6; 7 studies) favoring ultra-sound guided injection.

The dose of corticosteroid seems to make little difference, particularly at long term. Dammers et al. compared 20 mg, 40 mg, and 60 mg; Atroshi et al. compared 40 mg and 80 mg and Habib et al. compared 12 mg and 35 mg of methylprednisolone [34,35,36]. Roghani injected 80 mg versus 40 mg triamcinolone [37]; and O’Gradaigh et al. [38], 25 mg versus 100 mg of hydrocortisone [38]. None found clinically relevant short term or long-term benefits with the higher dose.

### 4.3. Do Corticosteroid Injections Work?

Several RCTs have compared corticosteroid injection with either placebo injection (saline or local anesthetic) or with no treatment [34,38,39,40,41,42,43,44]. Pooling data from three trials at short term (2 to 4 weeks) showed that the probability of improvement was 0.71 with corticosteroid and 0.31 with placebo corresponding with a relative risk of 2.2 (95% CI 1.6 to 3; 3 studies; 210 participants) [39,40,43]. The BCTQ symptom severity score favors corticosteroid injection in most studies. Pooling the short-term results suggests a clinically unimportant benefit of 0.3 points (95% CI 0.08 to 0.5; 3 studies; 220 participants) [34,42,43]. At long term (six months or more), the effect of corticosteroids seems to disappear in all studies [34,37,41,42].

While corticosteroid injection slightly decreases the risk of surgery during the first year, it may not decrease the rate of surgery when the follow-up continues for years. Atroshi et al. reported a surgery rate of 56/74 (76%) with corticosteroid versus 34/37 (92%) with placebo at one year (while participants were blinded to allocation). At five-year follow-up 65/74 (88%) of the participants treated with corticosteroid versus 36/37 (97%) with placebo had had surgery [34].

This high surgery rates suggest that corticosteroid injections can only be used to buy time, but the rates may vary based on population and may not be generalizable. Although not curative, the time gain allows commencement of other interventions.

## 5. Platelet-Rich Plasma (PRP) Injection

### 5.1. How the Intervention Might Work

Platelets release various growth factors and other active proteins at the injury site to promote healing of the injured tissue. PRP is a concentrate of plasma and platelets. It is widely used in many musculoskeletal indications, although limited evidence supports its ability to improve symptoms effectively [45,46,47]. Regarding nerves, animal models provide evidence for peripheral nerve injuries but the evidence in compression neuropathy is limited. When growth factors or PRP is injected into the nerve repair site, it seems to promote axonal regeneration, stimulate angiogenesis, and it has been hypothesized that it may also decrease fibrosis [48]. Since ischemia and subsequent fibrosis are elemental components of peripheral compression neuropathies, PRP could, in theory, improve the function of the compressed nerve.

### 5.2. What Is the Optimal Treatment Approach?

The lack of regulation of autologous blood products has allowed marketing and clinical use without normal early phases of drug development. Therefore, the optimal doses and preparation methods are unclear. A wide variety of preparation kits for autologous blood products are currently available and the compositions of the products may vary considerably [49].

### 5.3. Does PRP Work?

Evidence from two small placebo-controlled trials with methodological shortcomings indicates that the benefits of PRP may be clinically unimportant, but more evidence from large scale rigorous RCTs is needed to make firm conclusions [50,51]. Both studies used a single injection of PRP and blinded the participants, but it is unclear whether the allocation concealment was truly secured in either study.

Malahias et al. measured success by >25% improvement in the short version of Disabilities of Arm, Hand and Shoulder score (quick-DASH). They found no difference at one month, but at three months 20/26 had improved in pain with PRP versus 8/24 in the saline group. Mean differences were not reported, limiting the interpretation of results. The difference corresponds with a relative risk of 2.3 (95% CI 1.3 to 4.2). Chen et al. found 0.2 to 0.4 points (clinically unimportant) benefit in the BCTQ symptom severity score for PRP compared with placebo. Nerve conduction velocities were comparable between the groups; thus, the biological rationale of curative effects of PRP do not acquire support from this study.

## 6. Kinesio Tape

### 6.1. How the Intervention Might Work

Kinesio tape is an elastic tape applied to the skin allowing motion of the underlying joints. It is commonly used to treat various musculoskeletal symptoms. Its mechanism of action is largely unclear, but the hypothesis is that the tape could deform and stimulate large-fiber cutaneous mechanoreceptors. This may inhibit nociceptive impulses in the spinal column and decrease pain. Thus, in CTS, Kinesio taping likely modifies symptoms rather than intervenes with the pathological processes within the nerve.

### 6.2. Does Kinesio Tape Work?

Based on data from two sham-controlled trials with 77 participants (randomized to either sham or real Kinesio taping), Kinesio taping seems to provide little or no benefits in people with CTS [25,52]. Krause et al. compared real Kinesio taping with sham tape placed over the spine of scapula and did not find benefits in pain, BCTQ symptom severity score, or BCTQ functional scale [52]. Geler Kulcu et al. used Kinesio tape with tension as the active treatment and without tension as the sham treatment. They found no evidence of a clinically important difference between Kinesio taping and sham Kinesio taping.

## 7. Neurodynamic Techniques

### 7.1. How Neurodynamic Techniques Might Work

Experimental animal models of compression neuropathy suggest that compression reduces blood flow of the nerve leading to oedema, inflammation and subsequent intra-neural fibrosis and demyelination. This process ultimately causes thickening of the connective tissue around and inside the median nerve [32]. This fibrosis decreases the elasticity and gliding of the median nerve during wrist or finger motion possibly contributing to dynamic compression during hand activities [53].

Neurodynamic techniques aim to apply tension and/or cause gliding of the median nerve by mobilizing the joints and tissues of the upper extremity including the cervical spine. A systematic review found evidence that various neurodynamic maneuvers cause longitudinal and transverse excursion as well as changes in nerve diameter in vivo, but any quantifiable effects on nerve strain have not been reported [54]. The clinical effects are hypothesized to arise due to changes in the nerve excursion, intraneural oedema reduction, anti-inflammatory changes, activation of endogenous analgesic neural pathways and desensitization to mechanical loading through adaptation and habituation [54,55,56]. In asymptomatic people, neurodynamic techniques seem to cause a decrease in pain threshold compared with placebo maneuvers [57,58].

### 7.2. The Optimal Treatment Strategy

It is currently unclear if tensioning or gliding maneuvers (or some combination of them) provide clinically superior effects, or if some dosage is optimal [55]. Cadaveric and in vivo studies suggest that neural tissue excursion is greater with sliding techniques compared to tensioning maneuvers. On the other hand, the strain on the median nerve increases with tensioning techniques compared with movements that promote neural sliding. A combination of movements of adjacent joints seem to promote neural excursion more than single-joint movements [59,60].

### 7.3. Do Neurodynamic Techniques Work?

Limited evidence suggests that neurodynamic techniques could offer patient-important benefits in CTS, but there is a need for a large rigorous trial to improve the certainty of the evidence. Main limitations are the inconsistency between findings and risk of bias in the RCTs using a placebo control.

A recent systematic review assessed the effect of neurodynamic techniques and found no clinically important benefit in pain or the BCTQ symptom severity score when they pooled data from trials with various control treatments [61]. Two sham-controlled trials were included in this review. Bialosky et al., (*n* = 40) found no difference in pain or the DASH score between the sham and real treatment. However, study by Wolny and Linek (*n* = 180) found clinically important benefit for the neurodynamic group in the BCTQ symptom severity score (mean difference 1.22 points; 95% 1.1 to 1.4 points), BCTQ functional scale (mean difference 0.9; 0.7 to 1.1) and pain (mean difference 4.53 points; 95% CI 4.17 to 4.79) [62]. The results from the latter study are profoundly inconsistent with the other trials included in the systematic review [61]. Wolny and Linek have also published an open-label trial (*n* = 122) comparing neurodynamic techniques with no intervention (unblinded participants) showing essentially similar results as the sham-controlled trial [63]. The blinded study found no benefits in nerve conduction velocities indicating that the improvement could relate largely to other effects than improvements in the median nerve function (e.g., reduced mechanosensitivity) [62]. However, in the open-label trial, the nerve conduction velocity was 12 m/s higher in the experimental group [63]. 

Neurodynamic techniques have also demonstrated similar symptom relief as surgery, and only 15% of the participants had had surgery in the neurodynamic group at four years follow-up suggesting that the effects may carry over a long period [64,65,66].

In summary, the evidence supporting clinically important efficacy with neurodynamic treatments comes mostly from one research group, and other investigators have not been able to replicate such positive findings so far or have investigated neurodynamic treatments as part of a multimodal approach [67]. The results seem encouraging and symptom relief may last for long periods, but more research is needed in this area.

## 8. Gabapentin

### 8.1. How the Intervention Might Work

Gabapentin is an anti-convulsive drug used to treat pain. Moderate certainty evidence suggests that it improves neuropathic pain in diabetic neuropathy and post-herpetic neuralgia [68]. Thus, it could alleviate symptoms in CTS, although there is no rationale for how it could improve the function of the median nerve itself.

### 8.2. Does Gabapentin Work?

The current evidence from two efficacy RCTs does not support the use of gabapentin to treat CTS symptoms. Eftekharsadat et al. [69], (*n* = 90) had two active groups (100 mg a day and 300 mg a day) and an unblinded control group [69]. A placebo-controlled study (*n* = 140) used a starting dose of 300 mg a day increasing to the target dose of 900 mg a day. The placebo-controlled study found no benefit for gabapentin at two or eight weeks [70]. The most common side effect was dizziness (40% with gabapentin versus 28% with placebo), but there were no significant differences in the total number of adverse effects. However, the study was not powered to detect differences in adverse effects. The unblinded trial found a small but clinically unimportant benefit for gabapentin, but this could relate to bias arising from lack of blinding [69]. We did not identify any studies assessing the efficacy of pregabalin in this population, and as long as anticonvulsants do not show patient-important benefits with acceptable side effects, clinical use cannot be recommended.

## 9. Ultrasound

### 9.1. How Ultrasound Might Work

Ultrasound delivers energy to deep tissues increasing the temperature of the tissue. This may improve the regeneration of crushed nerve, and affect the conduction velocity, but the exact physiological mechanisms are unclear [71,72].

### 9.2. What Is the Optimal Strategy?

One small trial (*n* = 18 in three groups; placebo, 1.5 W/cm^2^ and 0.8 W/cm^2^) found no difference between the two active groups but due to the small number of participants, the treatment effect estimates are too imprecise to draw firm conclusions [73].

### 9.3. Does Ultrasound Work?

Ultrasound may modify the symptoms short term, but there are no large-scale high-quality trials available at the moment. It is unlikely that the effects would be sustained over a long period due to a lack of mechanism of action, and we found no efficacy data beyond three months.

Three placebo-controlled studies suggest that ultrasound five days a week for 2–3 weeks may have a small effect on pain and symptoms measured at 2–6 weeks [74,75]. One study followed the participants for 12 weeks and the effects had disappeared at the last follow-up [74]. One trial found no difference in pain or frequency of sleep disturbances between placebo, 1.5 W/cm^2^, and 0.8 W/cm^2^ delivered five days a week for two weeks [73].

## 10. Extracorporeal Shock Wave Therapy (ESWT)

### 10.1. How ESWT Might Work

In ESWT, the median nerve and surrounding tissues are exposed to repetitive acoustic pulses with high peak pressure followed by a pulse of negative pressure. The pulses can be radial or focused. Rat models suggest that ESWT may improve nerve regeneration and pain [76,77]. On the other hand, it may also damage the myelin sheath of the nerve [78]. Thus, ESWT could, in theory, modify the disease and symptoms, but there is currently no plausible explanation of how ESWT would improve the median nerve long term function.

### 10.2. Does ESWT Work?

At least three small sham-controlled studies have evaluated the efficacy of ESWT. The results indicate that ESWT may improve symptoms at one month, but at three months the results are inconclusive due to large inconsistency between the studies and no long-term follow-up exists.

At 1 month, two studies (from the same research group) found a benefit in the BCTQ symptoms severity score [79,80]. Pooling the data, the mean difference in the BCTQ was −0.5 points (95% CI −0.3 to −0.8; 100 participants) indicating a statistically significant and potentially clinically important benefit for ESWT. At three months, the results were inconsistent: one study found a large benefit of 1.4 points in the BCTQ symptoms severity score [81], while another [80] found virtually no benefit. It is unlikely that the differences in the protocols (one session/week for three versus four weeks) explain the large heterogeneity. Pooling data from two trials with consistent results, the benefit was likely clinically unimportant (0.3; 95% CI −0.7 to 0.08; 100 participants) but the confidence intervals do not exclude clinically important benefit [79,80]. The large benefit observed in the Vahdatpour et al. study was still present at six months. The benefit in pain was small and likely clinically unimportant (0.6 and 0.9 points on a 0 to 10 scale) in both studies that measured it [79,80].

It is unclear if ESWT can improve the neurophysiological parameters. Wu et al. and Ke et al. did not find a difference in the nerve conduction velocity, but Vahdatpour et al. found a significant difference in sensory latency.

To summarize, three small trials show conflicting results and indicate that ESWT may modify CTS symptoms by yet unknown mechanisms short-term, but there is no data beyond six months and biological rationale for long-term effects is lacking. There is a need for a large-scale rigorous sham-controlled trial to assess the efficacy before a firmer conclusion can be made.

## 11. Summary

Optimally, treatment not only has a plausible theory explaining how it exerts its effect, but it also works (shows patient-important benefits) in a rigorous large placebo-controlled trial(s), delivers relevant benefits also in a pragmatic clinical setting with a diverse population and care providers (effectiveness) and, finally, is worth using (cost-effectiveness). None of the treatment modalities in this review fulfils all these criteria. While most treatments showed small benefits in individual trials the overall results are often inconsistent between studies, and most studies have serious methodological shortcomings.

In some studies, even potentially clinically important benefits were observed but never in several independent high-quality studies. Sufficiently powered, rigorous placebo-controlled trials seem to be scarce in this area and therefore it is difficult to conclude which treatments, if any, are worth widespread clinical use. Although most treatments turned out to provide little benefits, these benefits could be additive and multimodal approaches should be assesses as carried out by Lewis et al. [67].

The treatment decisions need to take account of the individual needs as well as local resources. As long as high-certainty evidence of efficacy does not exist, comparative effectiveness studies (i.e., comparing two interventions with unknown effects) are less informative and therefore the research in this area should focus on establishing the efficacy of each treatment first.

## Data Availability

No new data were created or analyzed in this study. Data sharing is not applicable to this article.

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
