# Peer review of "Update on Efficacy of Conservative Treatments for Carpal Tunnel Syndrome"

_jcm, 2022, doi:10.3390/jcm11040950_

Round 1
Reviewer 1 Report
The papaer is a review about the Coseravtive Treatments of carpal tunnel syndrome.
The title is misleading as I did not find to much information about the effiency, especilally in comparison to the therapeutiucal gold standard, which is carpal tunnel release.
The paper itsself is well writte and readable. It's a good overview about existing conservative treatment options. Howesver, as the authors state temselves the study does nit really add much to the scientific data, although I know this was not the primary aim of the paper. Still, I miss a proper discussion of the topic.
Some remarks:
- affiliation: please check Aff 1
- kewords: corticoids are metioned twice
- Line 49-62: Here should at least a section about risks and complications should be added
Author Response
Reviewer comment. The papaer is a review about the Coseravtive Treatments of carpal tunnel syndrome.
The title is misleading as I did not find to much information about the effiency, especilally in comparison to the therapeutiucal gold standard, which is carpal tunnel release.
Authors reply: The title is update on efficacy of conservative treatments for carpal tunnel syndrome. Efficacy can not be assessed by comparing conservative treatments with carpal tunnel surgery, particularly because the efficacy of carpal tunnel surgery is poorly studied. Although enticing, we refrain here to compare agains surgery since 1) the review would be such a long if we included this comparison with all modalities; and 2) some of the authors of this review are also authors of Cochrane review on this particular matter (submitted but not published) that will be out sooner or later this year.
We have explained what we mean by efficacy in the introduction:
“Several RCTs have assessed the effect of various non-operative treatments in people with CTS, but often studies compare one treatment with another not addressing the question whether the interventions are better than doing nothing, i.e. their efficacy. In this narrative review, we will provide up-to-date data regarding the efficacy of orthoses, corticosteroid injections, platelet-rich plasma injections, Kinesio taping, neurodynamic techniques, gabapentin, therapeutic ultrasound, and ESWT interventions for CTS.”
Reviewer comment: The paper itsself is well writte and readable. It's a good overview about existing conservative treatment options. Howesver, as the authors state temselves the study does nit really add much to the scientific data, although I know this was not the primary aim of the paper. Still, I miss a proper discussion of the topic.
Authors reply: Thank you, we agree that we did not create much new knowledge since this was a narrative review synthesising the existing evidence. The title was given by the guest editor. We do discuss the limitations of current evidence on efficacy in the concluding section which we believe is the most important synthesis and take home message of this review and probably something not discussed elsewhere
Reviewer comment: Some remarks:
-
affiliation: please check Aff 1
-
kewords: corticoids are metioned twice
-
Line 49-62: Here should at least a section about risks and complications should be added
Authors reply: Thank you. We checked the affiliation and corrected the repetition in the keywords. We are sorry but we do not understand what is meant by risk and complications. Doing nothing cannot cause complications beyond the condition getting worse and the following paragraph (lines 62-65) reports the estimates the risk of the condition getting worse.
Reviewer 2 Report
I would like to thank the authors and Editors for inviting me to review this paper. This is a narrative review that summarises the efficacy of non surgical interventions for patients with carpal tunnel syndrome. This review has been carefully written and it is clear that the authors critically appraised the literature. Of course, a systematic review approach rather than a narrative approach would have increased the certainty and confidence in their findings, but I understand if that was not the purpose of this review. For instance, there are some recent large-scale trials missing from this review (e.g., Chesterton et al 2018, Lewis et al 2020 etc).
Below, I have several comments that I think would help improve the manuscript further.
Introduction, line 31: The authors nicely introduce risk factors for CTS. I suggest to also include genetic risk factors (see Radecki 1994, Wiberg et al 2019) and rheumatoid arthritis (see Geoghegan et al 2004, Karadag et al 2012) for completeness.
Orthoses line 86: the sentence “synteshsis of the evidence suggests that the benefit from splinting at short term may be little” suggests that this may not be the case for long term. But in fact studies currently point towards evidence for short term rather than long term benefit, even if small. I suggest the authors reword this for clarity.
PRP lines 174 onward. It is mentioned that a systematic review exists on PRP for CTS, however the results are not reported. Please include the results. It also remains unclear why some studies are singled out (e.g., Refes 47, 48 and Malahias): were these published after the systematic review was completed? Or were they not included in the systematic review?
Kinesiotape lines 194 onwards: please provide a reference for the hypothesis that the mechanism of action is through activation of mechanoreceptors in skin.
Lines 255: the sentence that only 14% of participants had surgery in the neurodynamic group is somewhat unclear. Can you please clarify this sentence so that it is clear to the reader what the neurodynamic treatment was compared to.
I agree with the authors about the caveat about certain studies on neurodynamic treatments for CTS. However in addition to the studies by Wolny, the studies by Fernandez-de-las-Penas also point towards benefit of these treatments, so I suggest that the sentence on lines 259-261 may not accurately reflect this issue. There is also some recent work from Lewis et al 2020, which used a combination treatment including neurodynamic exercise and also found a beneficial effect related to surgery prevention. I suggest the authors clarify these, but I agree that a notion of caution and interpretation of these studies is warranted.
I agree with the call for large scale trials comparing single interventions to placebo/wait and see. Would it also be fair to suggest that combination treatments should be investigated to examine whether the relatively small effects of single modalities have a cumulative effect once combined?
Author Response
Reviewer comment: I would like to thank the authors and Editors for inviting me to review this paper. This is a narrative review that summarises the efficacy of non surgical interventions for patients with carpal tunnel syndrome. This review has been carefully written and it is clear that the authors critically appraised the literature. Of course, a systematic review approach rather than a narrative approach would have increased the certainty and confidence in their findings, but I understand if that was not the purpose of this review. For instance, there are some recent large-scale trials missing from this review (e.g., Chesterton et al 2018, Lewis et al 2020 etc).
Authors reply: Thank you. We agree that systematic review is more rigorous method but this was the requested topic and as a very general and broad topic, probably not feasible for systematic review. We did, however, perform a systematic search to identify relevant studies.
Reviewer comment: Below, I have several comments that I think would help improve the manuscript further.
Introduction, line 31: The authors nicely introduce risk factors for CTS. I suggest to also include genetic risk factors (see Radecki 1994, Wiberg et al 2019) and rheumatoid arthritis (see Geoghegan et al 2004, Karadag et al 2012) for completeness.
Authors reply: Thank you. We have now included these risk factors and recommended references
Reviewer comment: Orthoses line 86: the sentence “synteshsis of the evidence suggests that the benefit from splinting at short term may be little” suggests that this may not be the case for long term. But in fact studies currently point towards evidence for short term rather than long term benefit, even if small. I suggest the authors reword this for clarity.
Authors reply: Thank you for pointing this out. It is exactly so – benefits in long term are even more uncertain. We have revised the text as: “However, shortcomings in methods, unexplained heterogeneity between studies and small sample sizes (imprecise treatment effect estimates) limit the interpretation, and particularly long term benefits are unclear since limited data exist beyond six months. “
Reviewer comment: PRP lines 174 onward. It is mentioned that a systematic review exists on PRP for CTS, however the results are not reported. Please include the results. It also remains unclear why some studies are singled out (e.g., Refes 47, 48 and Malahias): were these published after the systematic review was completed? Or were they not included in the systematic review?
Authors reply: Malahias was singled out from the systematic review since that was the only study in which the authors compared PRP with placebo. Due to our scope (efficacy), we have not included other comparisons from that particular systematic review. We have decided to remove the references to the systematic review because we only want to include placebo (or no treatment) comparisons and now it created confusion.
Reviewer comment: Kinesiotape lines 194 onwards: please provide a reference for the hypothesis that the mechanism of action is through activation of mechanoreceptors in skin.
Authors reply: Thank you. We think hypotheses do not necessarily need a reference when no evidence supports them. We revised the text slightly to further emphasize that it is just a theory
Reviewer comment: Lines 255: the sentence that only 14% of participants had surgery in the neurodynamic group is somewhat unclear. Can you please clarify this sentence so that it is clear to the reader what the neurodynamic treatment was compared to.
Authors reply: This (15% – there was a typo) is in absolute terms, I.e. 15% of the participants in the neurodynamic group had had surgery by 4 years. The trial compared NDT with surgery but surgery has nothing to do with the absolute rate and we were not interested in differences between surgery and NDT. We have now changed the number to 15% and removed word “furthermore” from the sentence.
Reviewer comment: I agree with the authors about the caveat about certain studies on neurodynamic treatments for CTS. However in addition to the studies by Wolny, the studies by Fernandez-de-las-Penas also point towards benefit of these treatments, so I suggest that the sentence on lines 259-261 may not accurately reflect this issue.
Authors reply: We agree that the 3 trials from the Spanish group (showing equal outcomes as surgery) point towards benefits. However, they are not efficacy studies and efficacy of surgery is still largely unclear and may actually be quite small (Gerritsen et al 2002 ; Jarvik et al. 2009, Verdugo 2008).
Our statement was specifically focused on clinically important efficacy ( if benefits are patient important) and thus we tried to reflect the evidence in this respect. We also state that the results are encouraging. Very serious inconsistency of results (between studies) and shortcomings in the methods clearly indicate that more research is needed. We have now revised this to reflect the finding from Lewis 2020 and it reads: “In summary, the evidence supporting clinically important efficacy with neurodynamic treatments comes mostly from one research group, and other investigators have not been able to replicate such positive findings so far or have investigated neurodynamic treatments as part of a multimodal approach. The results seem encouraging and symptom relief may last for long periods, but more research is needed in this area.”
Reviewer comment: There is also some recent work from Lewis et al 2020, which used a combination treatment including neurodynamic exercise and also found a beneficial effect related to surgery prevention. I suggest the authors clarify these, but I agree that a notion of caution and interpretation of these studies is warranted.
I agree with the call for large scale trials comparing single interventions to placebo/wait and see. Would it also be fair to suggest that combination treatments should be investigated to examine whether the relatively small effects of single modalities have a cumulative effect once combined?
Authors reply: Thank you. This is a very good notion that we overlooked in the concluding section. We have now added a sentence:
“Although most treatments turned out to provide little benefits, these benefits could be additive and multimodal approaches should be assesses as done by Lewis et.al. ”
PRP lines 174 onward. It is mentioned that a systematic review exists on PRP for CTS, however the results are not reported. Please include the results. It also remains unclear why some studies are singled out (e.g., Refes 47, 48 and Malahias): were these published after the systematic review was completed? Or were they not included in the systematic review?
Authors reply: Malahias was singled out from the systematic review since that was the only study in which the authors compared PRP with placebo. Due to our scope (efficacy), we have not included other comparisons from that particular systematic review. We have decided to remove the systematic review because we only want to include placebo (or no treatment) comparisons and now it created confusion.
Kinesiotape lines 194 onwards: please provide a reference for the hypothesis that the mechanism of action is through activation of mechanoreceptors in skin.
Authors reply: Thank you. We think hypotheses do not need references to support them. Anyone is free to propose hypotheses that are falsifiable. Because there is no evidence to support this hypothesis, we added no reference.
Lines 255: the sentence that only 14% of participants had surgery in the neurodynamic group is somewhat unclear. Can you please clarify this sentence so that it is clear to the reader what the neurodynamic treatment was compared to.
Authors reply: This (15% – there was a typo) is in absolute terms, I.e. 15% of the participants in the neurodynamic group had had surgery by 4 years. The trial compared NDT with surgery but surgery has nothing to do with the absolute rate and we were not interested in differences between surgery and NDT. We have no changed the number to 15% and removed word “furthermore” from the sentence.
I agree with the authors about the caveat about certain studies on neurodynamic treatments for CTS. However in addition to the studies by Wolny, the studies by Fernandez-de-las-Penas also point towards benefit of these treatments, so I suggest that the sentence on lines 259-261 may not accurately reflect this issue.
Authors reply: We agree that the 3 studies from Spanish group (showing equal outcomes as surgery) point towards benefits. However, they are not efficacy studies and efficacy of surgery is still largely unclear and may actually be quite small (Gerritsen et al 2002 ; Jarvik et al. 2009, Verdugo 2008).
Our statement was specifically focused on clinically important efficacy (and that the benefits are patient important) and thus reflect the evidence well. We also state that the results are encouraging, but very serious inconsistency of results and shortcomings in the methods clearly indicate that more research is needed. We have now revised this to reflect the finding from Lewis 2020 and it reads as: “In summary, the evidence supporting clinically important efficacy with neurodynamic treatments comes mostly from one research group, and other investigators have not been able to replicate such positive findings so far or have investigated neurodynamic treatments as part of a multimodal approach. The results seem encouraging and symptom relief may last for long periods, but more research is needed in this area.”
There is also some recent work from Lewis et al 2020, which used a combination treatment including neurodynamic exercise and also found a beneficial effect related to surgery prevention. I suggest the authors clarify these, but I agree that a notion of caution and interpretation of these studies is warranted.
I agree with the call for large scale trials comparing single interventions to placebo/wait and see. Would it also be fair to suggest that combination treatments should be investigated to examine whether the relatively small effects of single modalities have a cumulative effect once combined?
Authors reply: Thank you. This is a good notion that we overlooked in the concluding section. We have now added a sentence:
“Although most treatments turned out to provide little benefits, these benefits could be additive and multimodal approaches should be assesses as done by Lewis et.al. ”
Round 2
Reviewer 1 Report
Thanks, Reply was satisfying for me. In my opinion ready for publication